# Prediction of Motor Failure Time Using An Artificial Neural Network

**DOI:** 10.3390/s19194342

**Published:** 2019-10-08

**Authors:** Gustavo Scalabrini Sampaio, Arnaldo Rabello de Aguiar Vallim Filho, Leilton Santos da Silva, Leandro Augusto da Silva

**Affiliations:** 1Postgraduate Program in Electrical Engineering and Computing, Mackenzie Presbyterian University, Rua da Consolação, 896, Prédio 30—Consolação, São Paulo 01302-907, Brazil; leandroaugusto.silva@mackenzie.br; 2Computer Science Dept., Mackenzie Presbyterian University, Rua da Consolação, 896, Prédio 31—Consolação, São Paulo 01302-907, Brazil; arnaldo.aguiar@mackenzie.br; 3EMAE—Metropolitan Company of Water & Energy, Avenida Nossa Senhora do Sabará, 5312—Vila Emir, São Paulo 04447-902, Brazil; leilton@emae.com.br

**Keywords:** predictive maintenance, condition-based maintenance, artificial neural network, vibratory analysis, smart industry, industry maintenance

## Abstract

Industry is constantly seeking ways to avoid corrective maintenance so as to reduce costs. Performing regular scheduled maintenance can help to mitigate this problem, but not necessarily in the most efficient way. In the context of condition-based maintenance, the main contributions of this work were to propose a methodology to treat and transform the collected data from a vibration system that simulated a motor and to build a dataset to train and test an Artificial Neural Network capable of predicting the future condition of the equipment, pointing out when a failure can happen. To achieve this goal, a device model was built to simulate typical motor vibrations, consisting of a computer cooler fan and several magnets. Measurements were made using an accelerometer, and the data were collected and processed to produce a structured dataset. The neural network training with this dataset converged quickly and stably, while the tests performed, *k*-fold cross-validation and model generalization, presented excellent performance. The same tests were performed with other machine learning techniques, to demonstrate the effectiveness of neural networks mainly in their generalizability. The results of the work confirm that it is possible to use neural networks to perform predictive tasks in relation to the conditions of industrial equipment. This is an important area of study that helps to support the growth of smart industries.

## 1. Introduction

The control, monitoring and maintenance of production line equipment are fundamental activities for the quality and performance of the productive process [1,2,3,4]. Sensors and actuators play an important role in the operation of various machines such as conveyor belts, generators, mixers, compressors, furnaces, welding machines, among others, so they must always be in proper working condition. To guarantee this, these machines are constantly monitored and two types of maintenance of their components, corrective and the scheduled, are performed. Corrective maintenance is performed in the case of a critical failure in the equipment and causes an unplanned downtime of the production line. Scheduled maintenance is performed periodically and equipment is checked and replaced, if necessary, in order to avoid unplanned downtime [2,5,6,7]. Although scheduled maintenance is less disruptive, both types have associated costs due to loss of production. To avoid these two types of maintenance, industry has begun to perform condition-based maintenance where predictive equipment status is used to plan a maintenance. Employing this method has become part of the smart industrial maintenance and results in fewer downtimes than scheduled maintenance, as it avoids unnecessary maintenance, and reduces corrective maintenance by anticipating possible equipment failures [2,4,7,8,9,10,11]. Machine learning techniques such as Artificial Neural Network (ANN), Regression Tree (RT), Random Forest (RF) and Support Vector Machine (SVM) are being used to perform regression and prediction tasks in various applications [10,12,13,14,15,16,17,18,19,20,21]. These techniques have enabled predictive condition systems or remaining useful lifetime systems to be developed that allow different types of production variables to be used [10,22,23,24].

The objectives of this work were: (1) to propose a methodology to generate a training dataset based on vibration measurements. This methodology includes the characterization of the dataset and definition of a way of calculating the failure time in vibrating systems by means of the amplitude and frequency data; (2) to train an ANN to be able to predict the failure time of an equipment. This prediction allows anticipating maintenance only before a failure on motor occurs, reducing the production line downtimes and the costs involved. A computer cooling fan, with an accelerometer coupled to it to measure its vibrations, was used to simulate the evolution of motor vibration and collect the data to train and test the ANN. The training dataset was generated considering the frequency spectrum of the vibrations, containing amplitude and frequency of a time interval. Each pair of amplitude and frequency measurements was associated with a failure time. The efficiency of the model training was demonstrated by comparative tests with RT, RF and SVM machine learning techniques. The tests performed were the *k*-fold cross-validation standard test and model generalization test, both of them using the Root Mean Square Error (RMSE) performance index [14,15].

This work is organized as follows: Section 2 presents the related works; Section 3 presents the methods applied in the development of the proposed system; Section 4 presents the predictive tests performed and the results of the performance index; and Section 5 presents the conclusions and proposals for future studies.

## 2. Related Works

In the context of asset management in industry, condition-based maintenance plays an important role in seeking to reduce unnecessary maintenance, reduce downtime and reduce costs involved in these two aspects. In general, this maintenance strategy includes fault diagnosis, fault prognosis and maintenance process optimization [8,9,11].

Fault Diagnosis Systems (FDSs), play the role of fault diagnosis, aiming to detect and identify faults, characterized when a behavior or system parameter is out of acceptable conditions [25,26,27]. This type of system was studied in both small and local applications [25] as well as larger systems [26,27]. FDSs can be classified into two main groups, those using model-based techniques and those using model-free techniques. The first group uses mathematical models of the monitored system, which describe the behavior of the real process. This model is used to compare the behavior of the real system with that described by the model. The second group uses machine learning techniques to learn the different states of the monitored system to identify and classify faults [26,27,28]. Ntalampiras [26] presents an FDS in electrical Smart Grids (SG), where a model-free method was used to detect and isolate faults in SG, using data from the physical layer of the monitored system.

In contrast to FDSs, which aim to detect and classify faults, fault prediction systems comply with failure prognosis aspect, which aims to predict the future behavior of equipment and determine their possible moments of failure, assisting in decision making on maintenance issues. Yildirim, Sun and Gebraeel [8,9] presented a framework for generating efficient maintenance planning based on predictive analytics using Bayesian prognostic techniques. This predictive analysis dynamically estimates the remaining life distribution of electric generators, allowing estimating the maintenance cost and the best time for maintenance to occur. Verbert, Schutter and Babuška [11] also presented the optimization of maintenance through efficient failure prediction, the work propose a multivariate multiple-model approach based on Wiener processes for modeling and predicting of equipment degradation behavior. The work clearly traces the dependencies between the processes of fault diagnosis, fault prognosis and maintenance optimization. In the fault prognosis context, ANNs are important tools, because they enable the implementation of the prediction task easily and accurately.

### 2.1. Data Predicting with ANN

ANNs are structures inspired by biological neurons and formed by simple units of processing, called neuron. The neurons are connected to each other, and for each connection a synaptic weight is given. The training phase of the ANN adjusts the synaptic weights of these connections, modeling the relation of inputs and outputs of the system. ANNs have the ability to model nonlinear and complex problems and are easy to implement, as numerous libraries for various programming languages are available. The high generalizability of this technique is also a highlight, as data outside the training set is admissible in the system [15,24,29].

One of the tasks that can be performed by a ANN is prediction. This task was used to assist in the control and monitoring of numerous variables in several areas. In medicine, ANNs were used to predict mortality risk and incidence of disease [29,30,31]. Hao et al. [31] used ANNs to predict chronic disease risk in patients. The system developed used as input structured and unstructured data about the patient’s health conditions. In the field of wind power, they were used to predict wind speed and the amount of power that can be generated [24,32,33,34]. A study by Li, Ren and Lee [34] presents the used ANNs to predict wind speed, with the objective of reducing the effect of the instability of this variable and increasing the efficiency of the generation of electrical energy. The system uses wind velocity as input, transforming this data into average wind speed and wind speed turbulence intensity. In the area of railway engineering, studies have predicted failure point in rail turnouts and the rates of wear of wheels and rails [35,36]. Shebani and Iwnicki [36] presented a system for predicting wear in train wheels and train tracks. The system was developed with the aim of reducing maintenance costs, improving passenger comfort and even avoiding accidents. To carry out this prediction, the authors used an ANN having as inputs several variables such as the characteristics of the wheels and rails, train speed, yaw angle, etc. Other areas have also been targets of prediction studies using ANNs such as turbine operation [10], education [37], entertainment [38] and production [39].

### 2.2. ANN in Engines Failure Prediction Systems

ANNs were employed in studies of engines to analyse vibrations and predict equipment failure [2,4,7,22,40,41,42]. A study by Plante, Nejadpak and Yang [7] showed that by observing the vibration of motors it was possible to identify those considered normal in the operation of the machinery and those related to failure. From this analysis, along with the data concepts associated with vibration, it was possible to detect and predict faults and allows predictive maintenance to be planned and helps to establish the remaining life of the equipment. For each type of failure, the authors presented a frequency spectrum of the collected vibrations, allowing the analysis of the vibration behavior to identify possible types of failure. The use of the frequency spectrum to perform the analyses was also employed in the present work, since with this spectrum it is possible to have strong, reliable signal data. Gongora et al. [2] presented the failure classification of induction motor bearings based on ANNs using the motor stator current as data input. The work describes motor maintenance, the costs involved and the types of failures. It highlights the main types of engine failure and concludes that engine failure prediction can be performed by non-invasive methods such as vibration measurement. Günnemann and Pfeffer [42] presented the classification of defects of a motor through the measurement of vibration in its operation. This classification was performed with an ANN, having as input data the frequency spectrum of the collected signal, the output being a binary representing a defective or non-defective motor.

The ANN proposed in the present study was developed from the information gathered in the articles mentioned above. The methodology, which involves data collection and treatment, ANN training and performance measures, is described in the next section.

## 3. Proposed Method

In real systems, the collection and processing of data used for machine learning training should be performed historically in relation to the point of interest, like an equipment failure. Knowing when a particular equipment failed, a preliminary dataset would be extracted containing sensors data information. This dataset should then be refined and enriched with signal growth information and the time remaining for the failure to occur. This work presents a methodology to generate a training dataset similar to the dataset of a possible real system, with the difference that the vibration signal growth rate and the estimated failure time were artificially generated, for a better control of the scenarios and testing of the ANN. This methodology has as much importance as the results achieved, allowing vibration behaviors to be simulated and the data collected used to produce the training and test datasets for machine learning. Figure 1 shows the overview of the proposed method. The process begins with data collection, derived from the measurements made by the accelerometer coupled to a cooler fan. This data was then processed to generate training dataset. Among the procedures performed with this data are the Fourrier transform of the vibration signal, the definition of signal growth rates and the calculation of motor failure time. After the data were processed, the ANN was trained to predict the motor failure time and validated by means of a performance index.

### 3.1. Data Collection and Data Processing

The device model used in this work, to collect data, was comprised of a computer cooling fan with small magnets fixed to its blades. Adding a second magnet to certain blades created a weight difference between those with only one magnet and those with two magnets. This weight difference generated vibrations during the rotation of the fan’s motor, allowing the vibration to be controlled in order to generate different vibration scenarios. A microcontroller, the Arduino UNO, was responsible for setting the motor speeds and performing the readings of the vibration values from the accelerometer [43].

An Akasa AK-FN059 12cm Viper cooling fan was used in the construction of the device model and an MMA8452Q accelerometer was used to measure vibration, attached to the cooling fan [44,45]. This accelerometer has 12 bits of resolution and communicates with the microcontroller through the $I^{2}C$ (Inter-Integrated Circuit) protocol. It was developed a software using Processing program language to collect the data from the serial port and store it in a text file [46,47]. Figure 2 shows the device model developed to simulate motor vibrations.

In order for the training dataset to cover different levels of vibration, three weight distribution configurations were done in the cooler blades. Figure 3 presents these configurations, where the color pairs represent the position where the weights were doubled to generate different vibration behaviors. For each of these configurations, 17 rotation speeds were set up, ranging from 20% to 100% of the cooler maximum speed at 5% intervals. The vibration measurements of each of these speeds was collected by the accelerometer at a frequency of 20 ms for 1 min, generating 3000 records per speed. Thus, in total, 153,000 vibration records were collected from the simulation model.

The vibration data was pre-processed to standardize the inputs and outputs used in the training dataset. When analyzing the behavior of the collected signals, it was observed that the vibrations had no harmonic behavior, i.e., the signal did not show constant amplitude and frequency. For the same window of measurement of the same rotation speed, different and non-standard data were collected. Therefore, it was necessary to define a single frequency and amplitude value for each of the axes in a measurement window. Once 1 min of data was collected for each motor speed of the cooler, represented by 3000 observations, a set of 50 observations of the dataset were defined as the measuring window, which represent 1 s of signal. Figure 4 shows an example of the difference in measurements taken at the same speed rotation of the fan at two different measuring windows.

The Fourier series and Fourier transform were used to describe the signals in the frequency domain, mapping the various frequencies and amplitudes of the signal. From this transformation it was possible to define a single amplitude and frequency value per axis for each measurement window [48]. For this purpose, the calculation of the Fourier transform using the Fast Fourier Transform function was implemented in R, with the aid of the ’spectral’ library, generating all pairs of amplitude and frequency of a measurement window. To generate a unique value, for each window, the Root Mean Square (RMS) value of the signal, or effective value, for the amplitude and frequency sets, was calculated by the following equation [49,50,51]:(1)$$x_{rms}=\sqrt{\frac{1}{n}\cdot \sum_{i=1}^{n}x_{i}^{2}}$$
where $x_{rms}$ represents the effective value of the amplitude or frequency; $x_{i}$ represents each amplitude or frequency that compose the signal; and *n* the total number of amplitudes or frequencys that compose the signal.

A new dataset containing 3060 records of pairs of amplitude and frequency data for each vibration axis was generated. Figure 5 presents an example of the transformation performed on the vibration measurements graphically.

With the unique values calculated for each measurement window it was possible to establish the threshold values of amplitude and frequency needed to calculate the estimated failure time. The limit values, i.e., the maximum possible amplitude or frequency values before equipment failure, were defined by means of the analysis of the measurements made with the cooler rotating at the maximum speed. Thus, the highest values of each amplitude and frequency pair were found for each axis. The maximum amplitudes on the accelerometer measurement scale ([-8g,8g]) were 0.25 for the x axis and 0.7 for the y and z axes with a frequency of 18 Hz for all the 3 axes. The data were analyzed to remove attributes that would not be representative. The first attributes removed were the amplitude and frequency of the x axis, since the way the accelerometer was installed did not generate vibrations on that axis because it is the axis of height. The frequency attributes of the y and z axes were then removed. Although they generated values, the interval between the measured frequencies and the threshold was small and almost unchanged. Thus, only the amplitudes of the y and z axes were used for the training dataset. Figure 6 graphically presents the process of generating the amplitudes used in training dataset, where an amplitude value of the training dataset corresponds to the RMS value of a collected signal measurement window. In this figure, the amplitudes of y axis was represented.

To calculate the estimated failure time, three steps were performed in the dataset. The first was to establish vibration signal growth rates for the observations, doubling the dataset for the amount of desired growth rates. Three growth rates gr of the signal were set: 0.01, 0.02 and 0.05. In a real-time system this rate would be found by averaging the amplitude and frequency difference of two windows of sequential measurements, indicating how the vibration signal is growing. The second step was to calculate the time, in windows of measurement, so that each amplitude reached the threshold value according to the growth rate established for each amplitude of the axes. This calculation is given by:(2)$$FT_{x}=\frac{x_{lim}-x}{x\cdot gr}$$
where $FT_{x}$ represents the failure time considering one variable x (amplitude); $x_{lim}$ represents the threshold value of *x* and gr represents the growth rate of the signal. Finally, the third step was to calculate the average between the failure times, generating the expected equipment failure time. Thus, given the input $x=\left\{x_{1},x_{2},...,x_{n}\right\}$ and the signal growth rate gr, the expected output representing the estimated failure time $FT_{e}$ is given by:(3)$$FT_{e}=\frac{\sum_{i=1}^{n}\frac{x_{ilim}-x_{i}}{x_{i}\cdot gr}}{n}$$

This was a simple and efficient way to characterize the vibration dataset and can be performed at any time interval for the measurement windows. In the case of this work, the measurement window used was 1 s; however, it could have a duration of 1 min, 1 h or 1 day, with the result of calculating the equipment failure time given in the same units as the measuring window. The equation presented considers a linear growth rate, however exponential or logarithmic growth behaviors could be used depending on the desired or expected growth type. In a real dataset, this growth behavior would be discovered by the ANN.

### 3.2. Performance Index

All validations of the trainings performed in this work used the RMSE performance index. This performance index indicates the standard deviation of the difference between the estimated values and the values predicted [15]. The RMSE value was calculated with the following equation [14,15]:(4)$$RMSE=\sqrt{\frac{1}{n}\cdot \sum_{i=1}^{n}{\left(x_{i}^{\prime }-x_{i}\right)}^{2}}$$
where *n* represents the number of observations compared; $x_{i}^{\prime }$ the value of the i-th element of the predicted results vector; and $x_{i}$ the value of the i-th element of the test dataset estimated values vector.

### 3.3. ANN Training

At the end of dataset pre-processing, a new dataset was generated with 9180 observations and four attributes: amplitude of the y axis; amplitude of the z axis; signal growth rate; and estimated failure time. The first three attributes were used as input values for the training of the ANN, since the attribute estimated failure time was used as the output of the network.

The ANN class used to predict the failure time was the Multilayer Perceptron Neural Network (MLP). This ANN consists of an input layer, one or more hidden layers, and an output layer. The input signal propagates forward from layer to layer until it reaches the network output. The network training happens in a supervised way and, although there are others, the algorithm of backpropagation of error is the most common learning algorithm for this type of ANN. This architecture is flexible to parameterize inputs and outputs, in the case of this work the network allows configuring different sensors’ input signals. ANN neurons can be parameterized with nonlinear activation functions ensuring their use to continuous outputs. This feature allows to model complex systems that deal with nonlinear datasets, such as the relationship between vibration and equipment failure time addressed in this work. This class of ANN in conjunction with the backpropagation learning algorithm is simple to implement, efficient for large-scale problems, great generalizability and, as can be seen in Section 4 (Figure 7), depending on the dataset structure used for training, converges quickly and accurately, with the error tending asymptotically to 0. The backpropagantion learning algorithm can be used with an online update of synaptic weights; these updates occur with each new set of inputs in ANN training, so the model is refined by the training dataset observation number multiplied by number of iteractions (epochs), giving greater precision in the model. MLP neural network training with the backpropagation algorithm is also computationally efficient because computation is linear for all ANN synaptic weights, making the algorithm linear with respect to the number of *w* synaptic weights (O(w)). The literature shows that it is a consolidated model for use in time series prediction problems, considered in some cases as a benchmarking model [15,24,29].

The training of the ANN were performed in R with the help of the ’RSNNS’ library [51,52]. The training parameters were found empirically by trial and error, for each set of parameters the RMSE value was calculated and compared, the parameters that presented the smaller RMSE were: learning rate (specifies the gradient descent step width)—η=0.85; number of epochs (iterations)—Maxit= 50,000; number of hidden layers—nHidden=1; number of neurons in the hidden layer—$sizeHidden_{1}=25$. The ANN was trained with the backpropagation learning algorithm (learnFunc); the logistic function was used as the activation function of the hidden layer and the linear function was used as the activation function of the output layer (hiddenActFunc). The connections weight initialization between the neurons was performed in a random manner.

### 3.4. Comparing with Other Machine Learning Techniques

To establish the performance level of the model generated with ANN training, other classical machine learning techniques to perform the regression, estimation and prediction tasks were chosen for comparison. Regression Tree, Random Forest, and Support Vector Machine techniques, as well as ANNs, are widely used for classification and prediction problems in many application areas [10,14,15,16,17,18,19,20,21]. For this reason, they were selected to perform the same prediction task to which ANN was exposed.

Regression Tree is the name given to a Decision Tree used to perform the regression task. This structure consists of binary branches, where each node represents a decision regarding a simple comparison. These decision frameworks consider all training dataset inputs, modeling the relationship of these inputs to the expected output. The biggest advantage of this technique is the simplicity of tree construction, requiring little processing. Another advantage is the ease interpreting of the generated model. The generalizability of this technique can be reduced, as the tree structure strongly depends on the training dataset, so values outside of those considered during model training can lead to a large prediction error [10,12,16,49]. RT training was conducted in R with the help of the ’rpart’ library [51,52]. The choice of parameters for model training was performed in the same way as the ANN, empirically by trial and error. The parameters with the lowest RMSE value were: minimum observations on one node for a split attempt—minSplit=10; minimum number of observations in each tree leaf (node)—minBucket=round(minSplit/3); complexity parameter, removes worthless divisions—cp=0.0000001; maximum tree depth—maxDepth=30.

Random Forest corresponds to a set of tree predictors, each of these trees is dependent on the values of an independently taken random input vector with the same distribution for all trees in the forest. This structure combines several simple predictors, reducing complexity and improving performance when compared to individual tree models. For regression problems, RT is used as submodels of the RF structure. This technique is robust in the presence of outliers and noises in the training dataset and very stable to overfitting. Just as RT, RF may not perform well if the input data from the trained model is very different from the data presented during the training phase [10,13,18,19]. RF training was performed in R with the help of the ’randomForest’ library [51,52]. The choice of parameters happened empirically by trial and error. The parameters with the lowest RMSE value were: number of growing trees—nTree=500; number of variables sampled in each division—mTry=3; minimum size of terminal nodes—nodeSize=5; number of times data is exchanged per tree to verify the variable importance—nPerm=1.

Support Vector Machine is a popular machine learning technique used for classification, regression, prediction, and other problems. Nonlinear regression-driven SVMs, enabled by the intensive-loss function [53], perform during the training phase, basically, the mapping of input vector elements to high dimensional feature space using a nonlinear mapping process. This technique, like ANNs, is highly flexible from a training dataset entry point of view and tends to have a better generalization than tree-based techniques [10,15,17,21,23,49,53,54,55]. SVM training was conducted in R with the help of the ’svm’ function of library ’e1071’ [51,52]. The choice of parameters also happened empirically by trial and error. The parameters with the lowest RMSE value were: kernel used in training and prediction—kernel=radialbasis; cost of constraints violation—cost=10; parameter γ used in kernel calculation formula—γ=10; tolerance of termination criterion—tolerance=0001; parameter ϵ value of the insensitive-loss function—ϵ=0.0005.

The parameters presented were the ones that were varied to find the best RMSE performance index. There are other parameters for the training of each of these machine learning techniques; however, these other parameters were configured with the default values of the R libraries used. To consolidate and allow a better visualization of the used parameters, Table 1 presents the parameters by technique. Comparative tests between these techniques are presented in the next section.

## 4. Prediction Results

The models, using the machine learning techniques described in the previous section, were trained and tested. During ANN model training, the learning algorithm tends to reduce the output error interactively. Figure 7 shows the evolution of the weighted sum of squared error, for the k=1 folder. It can be seen from this figure that the error value tends asymptotically to 0 quickly and the error no longer varies after few training epochs. The other folders presented similar behavior in respect of error evolution during the training. The training phase involved *k*-fold cross-validation, to verify in a standardize way the performance of the trained model. The *k*-fold cross-validation enables the simulation of *k* data scenarios, where the test dataset is not use to train the model. After each folder training, the prediction model was tested with the test dataset and the values predicted and estimated are compared with the performance index [15]. By means of the RMSE calculation, using the estimated and predicted values, it was possible to measure if the generated model had precision and reflected the reality of the system being studied. In this work, the *k* value was set as 5. Also, the RMSE values can be used to compare the models performance. Figure 8 presents the graphs which demonstrate this comparison for the models generated in the k=1 folder, where the x axis represents the set of amplitudes of the signal and the signal growth rate (AY-AZ-GR) and y axis represents the failure time in seconds. For small amplitudes and small vibration signal growth rate the time until the equipment fails is longer; as the vibrations increase and vibration increases, the time until the equipment fails is shorter. The dataset used was generated considering a window of measurement of 1s which generated a failure time also in seconds, but this window could consider a measurement of 1day, which would generate a failure time in days, meaning that the measuring window is flexible. Table 2 shows the RMSE values for each model, validation folder and the average of this index.

All models converged during training and were able to deliver good performance results. The average RMSE for the ANN model was 0.0038 being ahead of the models trained with the RT and SVM techniques, the best result of *k*-fold cross-validation was presented by the RF technique, with an average RMSE value of 0.0026. Performance measurements of models, acquired through the test dataset, even with *k*-fold cross-validation, may present better values than a possible generalization in a real system. This scenario can happen if the dataset has values in its observations close to or even equal. In the dataset used in models training and testing, the amplitude and vibration growth rate values were doubled to simulate different scenarios of vibratory behavior generating optimal tests, since the test dataset can contain the same values for vibration amplitude in y and z as the training dataset. Thus, two new datasets were generated with different amplitude values and vibration signal growth rates in relation to the original training dataset, in order to attest to the generalizability of the trained model. Both datasets for generalization served as input to the models and the RMSE values of these new tests were calculated. Generalization dataset (a) was generated with the same vibration amplitude values as the models’ original training dataset, but with different growth rate values of 0.015, 0.03, and 0.04, totaling 9180 observations. Generalization dataset (b) was produced with the average amplitude values for each cooler motor rotation speed in the three weight settings of its blades, doubled by the vibration signal growth rates of 0.015, 0.03 and 0.04 totaling 153 observations. Comparative graphs between the estimated and predicted values of these two generalization datasets can be seen in Figure 9 and Figure 10, as well as numerically in Table 3. The values generated for these datasets did not exceed the maximum and minimum values of the models training dataset, in order not to impair the performance of tree-based techniques. It is possible to observe in these figures and by analyzing Table 3 that the prediction followed the same behavior as the one presented in the *k*-fold cross-validation, in a more accentuated way. Short term predictions were accurate for ANN, RT and RF techniques, however for medium and long term predictions ANN outperformed the other techniques. In general ANN had the best RMSE indexes for the generalization tests with 0.0313 for generalization (a) and 0.118 for the generalization (b). The values presented are low, proving the good generalizability of the model. The RT and RF models had a good generalization for short term predictions and, even with low RMSE indexes, SVM did not perform very well.

The results presented in tests with ANN prove that the objective of the work was achieved, and that a real-time system could be implemented to predict the failure time associated with vibration in motors using this machine learning technique. This system should have as input the same parameters used in the ANN training: vibration amplitude on the y axis; amplitude of vibration on the z axis; and vibration signal growth rate, calculated with the current vibration values and the values of the last measurement. The system would output the estimated failure time with good accuracy.

## 5. Conclusions

Increasingly, businesses seek more efficient ways to perform activities that directly impact their performance. In the case of the advent of smart industry, accurately predicting when any equipment or process may fail helps decision making regarding maintenance that needs to be performed, minimizing costs and reducing workload.

The objectives of this work and its main contributions were to propose a methodology to treat the collected vibration data from a device model, developed to simulate an actual system by measuring the vibrations of a computer cooler fan with an accelerometer, and to build a dataset to train a ANN capable of predicting when a failure can happen. The construction of this dataset involved the estimation of equipment failure time, achieved by a formula that considers the linear growth of the signal. The ANN class chosen to perform the prediction task was MLP, which is an easy to implement ANN with a good generalization index. The ANN model was compared in terms of RMSE performance index values with other machine learning techniques: Regression Tree; Random Forest; and Support Vector Machine. The results of the training and comparative tests were satisfactory, showed that the ANN model was superior to the other techniques. Generalization in short term predictions were equivalent between ANN, RT and RF techniques, however with the ANN model, medium and long term predictions were better. The success of the training and testing of the MLP neural network described in this work to predict motor failure shows that this technique could be applied in industrial condition-based maintenance. Despite the good performance of the ANN for the dataset built in this paper, MLP neural networks with backpropagation learning algorithm may have some weaknesses. The literature [15] points out a tendency of these techniques to converge slowly, besides allowing the occurrence of overfitting, where after a certain time of training the error curve, until close to 0, moves away from 0. Thus, some of these points can be explored in future work. Different classes of ANNs could be implemented and compared in order to observe which is the best model for the problem discussed in this paper. From an asset management point of view, it would be interesting to evaluate the possibility of jointly implementing fault diagnosis systems and fault prediction systems with ANNs, with the aim of building a unified system for maintenance planning in the industry. The proposed methodology could also be target of future works, considering real systems measurements and other variables such as temperature and pressure of the equipment, since the behavior of these variables are different and would influence the construction of fault prediction systems differently. The training dataset could be constructed based on real measurements or simulated with nonlinear estimates such as logarithmic or exponential.

It is important to highlight that research and development of techniques that helps decision making in the industry are essential for greater production efficiency and cost reduction. The development of low cost and easy implementation solutions are attractive to industry, which can materialize academic knowledge and development.

## Figures and Tables

**Figure 1 sensors-19-04342-f001:**
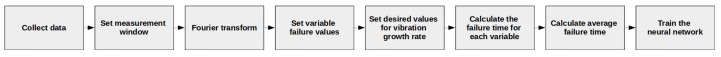
Data collection and pre-processing flow chart.

**Figure 2 sensors-19-04342-f002:**
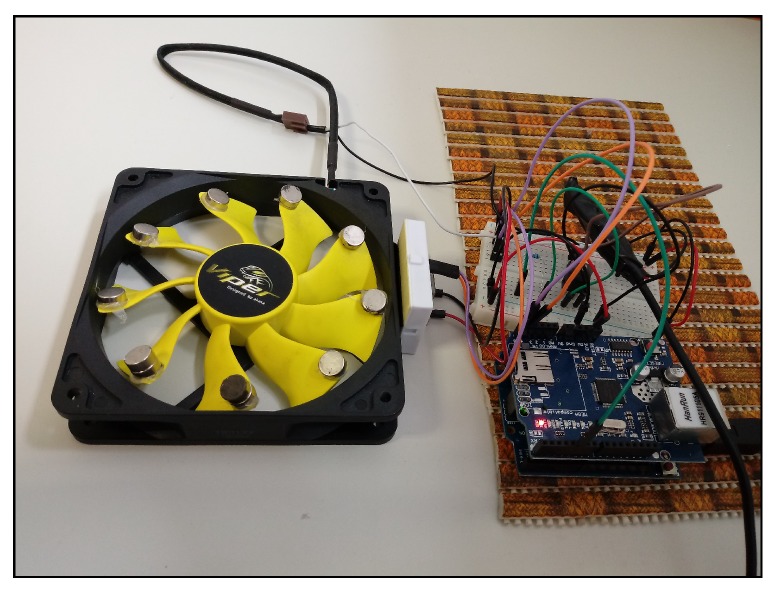
Device model developed to simulate vibrations in motors.

**Figure 3 sensors-19-04342-f003:**
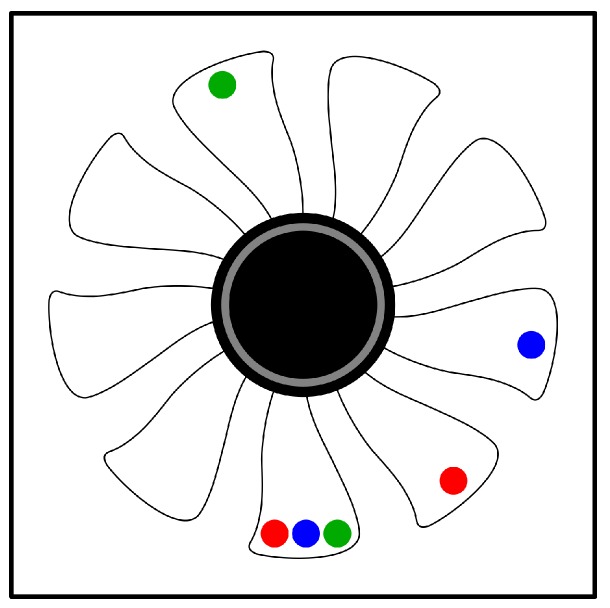
Weights distribution configurations between the cooler’s blades, performed to collect different vibration behaviors.

**Figure 4 sensors-19-04342-f004:**
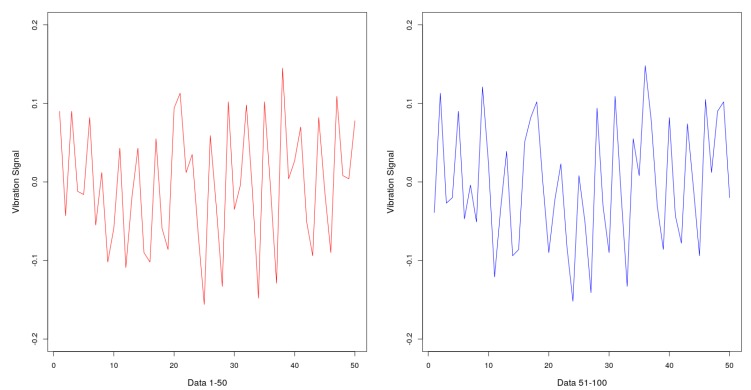
Vibration signals from two different and sequential measuring windows.

**Figure 5 sensors-19-04342-f005:**
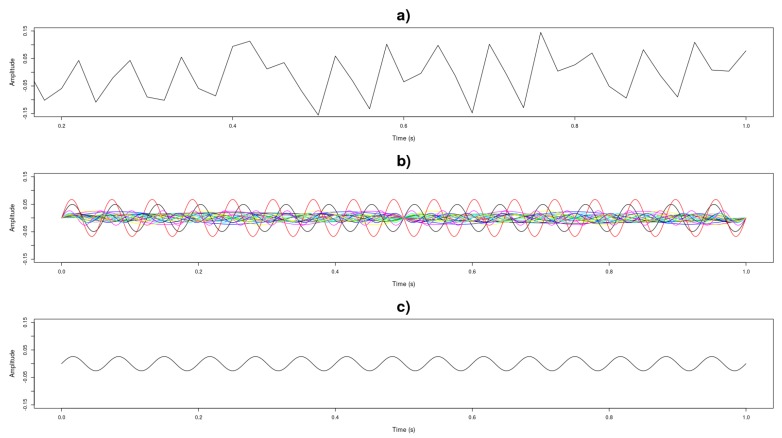
Process of simplification of measured vibration signal. (**a**) Vibration signal collected from a measuring window; (**b**) Application of the Fourrier Transform in the collected signal, generating all pairs of amplitude and frequency present in the signal; (**c**) Calculation of the RMS value of the amplitudes and frequencies, generating only one pair for each measurement window.

**Figure 6 sensors-19-04342-f006:**
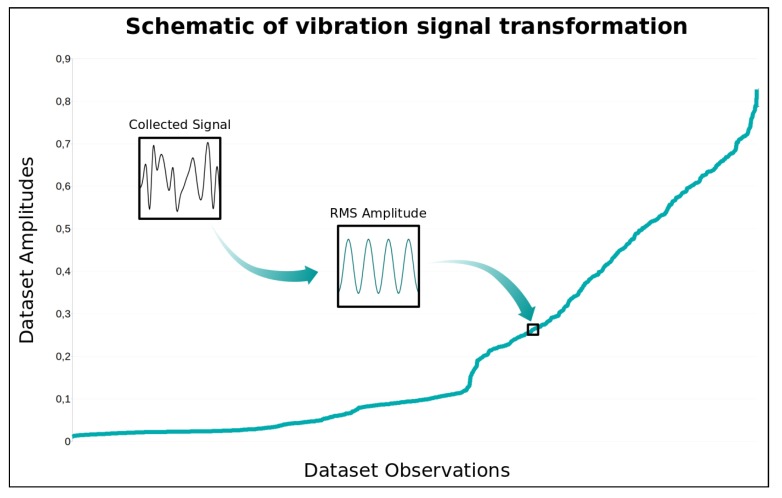
Schematic of vibration signal transformation to generate amplitude dataset (AY).

**Figure 7 sensors-19-04342-f007:**
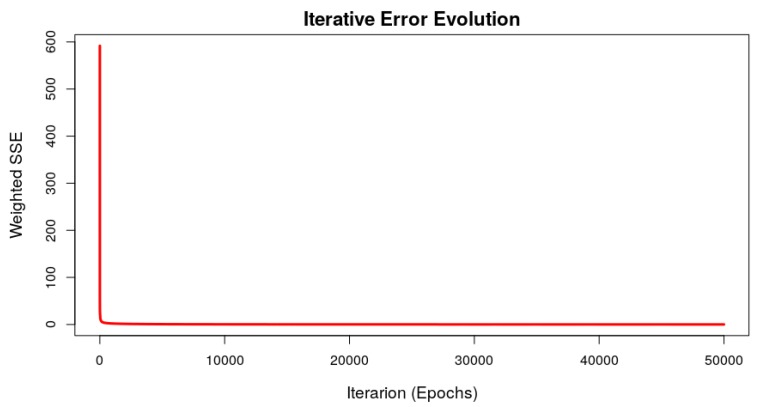
Iterative error evolution of the ANN training process (k=1).

**Figure 8 sensors-19-04342-f008:**
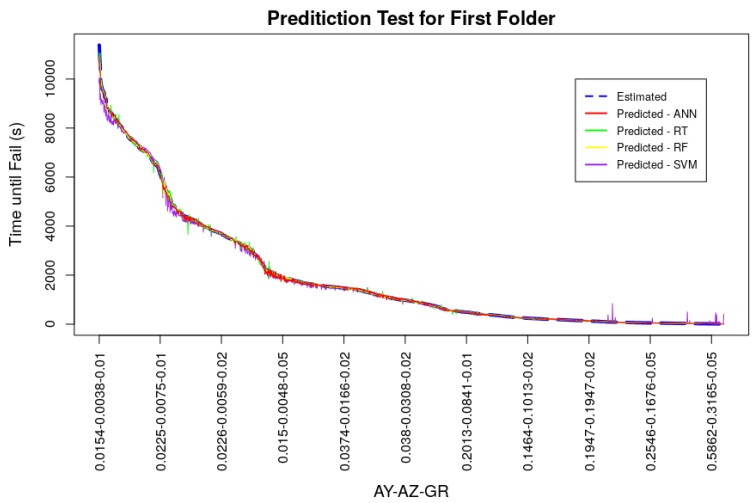
Results of the tests of the first folder (k=1) carried out with the machine learning techniques.

**Figure 9 sensors-19-04342-f009:**
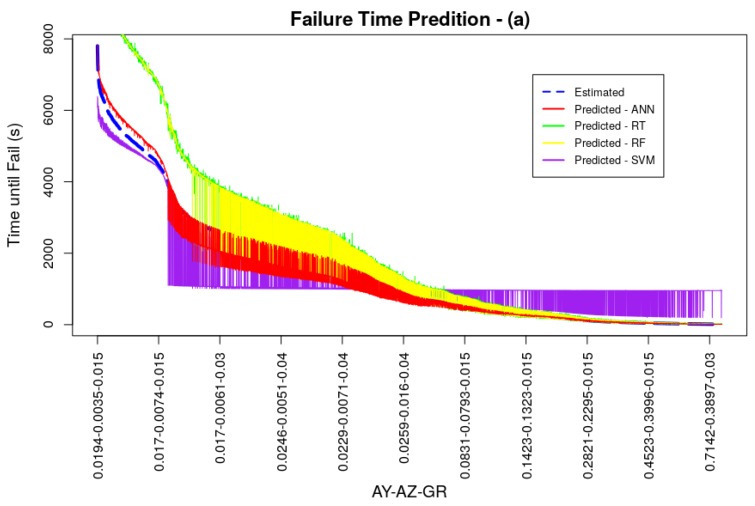
Generalization Test (a).

**Figure 10 sensors-19-04342-f010:**
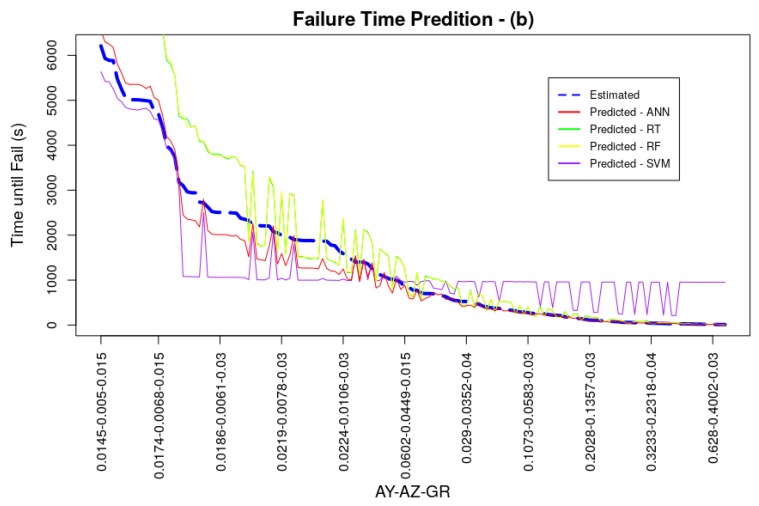
Generalization Test (b).

**Table 1 sensors-19-04342-t001:** Machine learning techniques parameters.

ANN		RF	
	nHidden=1		nTree=500
	$sizeHidden_{1}=25$		mTry=3
	η=0.85		nodeSize=5
	Maxit(epochs)= 50,000		nPerm=1
	learnFunc=Backpropagation		
	hiddenActFunc=Logistic		
**RT**		**SVM**	
	minSplit=10		kernel=radialbasis
	minBucket=round(minSplit/3)		cost=10
	cp=0.0000001		γ=10
	maxDepth=30		tolerance=0.0001
			ϵ=0.0005

**Table 2 sensors-19-04342-t002:** RMSE values of the folders used in the test of the machine learning techniques.

Folder	ANN	RT	RF	SVM
1	0.0039	0.0047	0.0025	0.0106
2	0.0035	0.0054	0.0035	0.0129
3	0.0028	0.0051	0.0022	0.0105
4	0.0041	0.0052	0.0024	0.0120
5	0.0049	0.0052	0.0026	0.0123
Average	0.0038	0.0051	0.0026	0.0117

**Table 3 sensors-19-04342-t003:** RMSE values of the generalization test of the machine learning techniques.

Gen.	ANN	RT	RF	SVM
a	0.0313	0.0922	0.0920	0.0696
b	0.1184	0.1417	0.1430	0.1237

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
