# Peer review of "Prediction of Motor Failure Time Using An Artificial Neural Network"

_sensors, 2019, doi:10.3390/s19194342_

Round 1

Reviewer 1 Report

The paper deals with an interesting application of neural networks for the prediction of motor failure time by measuring vibrations.

The paper topic is interesting and relevant due to the importance of predictive maintenance of the productive process.

The general lay-out of the paper can be improved: the description of the proposed methodology and examples are not clearly separated, making it hard to get a clear overview of the paper.

Some of the basic terms are not explained, ie. "vibration intensity"

Author Response

We appreciate your attention to our work and we understand that this process seeks to improve its quality. Follow our comments on the questions proposed:

RC: Reviewer Comment
AC: Authors Comment

RC: The general lay-out of the paper can be improved: the description of the proposed methodology and examples are not clearly separated, making it hard to get a clear overview of the paper.

AC: We have restructured section 3, which deals with the proposed methodology. First we present an overview of the process of generating the neural network training base and then detailing each of the steps. We hope that this way the methodology will be clearer to the reader. Please, see the highlighted text (lines 136-142).

RC: Some of the basic terms are not explained, ie. "vibration intensity"

AC: We try to avoid expressions like the above by removing or simplifying them. Please, if still some expression of difficult understanding let us know.

Reviewer 2 Report

The paper area is of high interest; however, the paper needs to be improved to reinforce its research contribution, and implemented on real industrial cases to demonstrate the immediate application to reality.

Research contribution needs to be mentioned in the abstract.

The authors should use “asset management concepts” in order to relate the application in a maintenance department, especially related to Condition-Based Maintenance policies, and about IPF interval, a literature review is missing in this matter.

Why do you select ANN? Why do not you consider other methods? It is not a new method in AM, and ANN has several critics as a black box because it’s difficult to explain its logic and weight adaptation. I recommend to compare with other intelligent techniques and with stochastic methods, showing performances in terms of RMSE, MAE, and MAPE, and using not only precision, accuracy, sensibility analysis and ROC (receiver operating characteristics).

Parameter initialization or optimization is key to affect the performances of ANN. The authors need to show the process and the optimized parameters in the experimental results.

Research contribution needs to be mentioned in Conclusions. Research limitations, and advantages and disadvantages of each technique need to be mentioned in Discussions.

Author Response

We appreciate your attention to our work and we understand that this process seeks to improve its quality. Follow our comments on the questions proposed:

RC: Reviewer Comment
AC: Authors Comment

RC: The paper area is of high interest; however, the paper needs to be improved to reinforce its research contribution, and implemented on real industrial cases to demonstrate the immediate application to reality.

AC: We try to emphasize more the contributions of the work in several passages. Please see the highlighted text (Abstract; lines 37-42; lines 329-339). The application in real systems could not be addressed in this work, however we intend to accomplish this task in future work. Please, see the highlighted text (lines 340-353).

RC: Research contribution needs to be mentioned in the abstract.

AC: We rewrote much of the abstract. Please, see if the contributions of the work are clearer.

RC: The authors should use “asset management concepts” in order to relate the application in a maintenance department, especially related to Condition-Based Maintenance policies, and about IPF interval, a literature review is missing in this matter.

AC: We include fault prediction in the context of asset management and condition-based maintenance. Please, see the highlighted text (Abstract; lines 54-79).

RC: Why do you select ANN? Why do not you consider other methods? It is not a new method in AM, and ANN has several critics as a black box because it’s difficult to explain its logic and weight adaptation. I recommend to compare with other intelligent techniques and with stochastic methods, showing performances in terms of RMSE, MAE, and MAPE, and using not only precision, accuracy, sensibility analysis and ROC (receiver operating characteristics).

AC: We include the justification of the technique used in 2 points in the text. Please, see the highlighted text (lines 81-87; lines 227-247). It was not possible to compare with different artificial intelligence techniques in this work, but we intend to make this comparison in future work when we apply the proposed methodology in a real system. Please, see the highlighted text (lines 340-353).

RC: Parameter initialization or optimization is key to affect the performances of ANN. The authors need to show the process and the optimized parameters in the experimental results.

AC: In this work the parameters for neural network training were performed empirically by trial and error. The parameters presented were the ones that presented the best performance of the prediction model. Please, see the highlighted text (lines 260-268).

RC: Research contribution needs to be mentioned in Conclusions. Research limitations, and advantages and disadvantages of each technique need to be mentioned in Discussions.

AC: We have tried to include and clarify these points in the conclusion section. Please, see the highlighted text (lines 329-357).

Reviewer 3 Report

In this work, the authors present the experience without the adequate detail of the most important parts; English Review: “The present work presents” (?) etc.; Abstract: I suggest to not considering the RMSE value. It is an information with no relevance. The convergence/stability during the training must be emphasized; Citations are missing in important parts of the text, for example, some (many) equations, “root mean square error, artificial neural network, support vector machine, decision tree, multilayer perceptron ANN, backpropagation algorithm etc.; I suggest to use benchmark publications, g., backpropagation algorithm [A]. P.J. Werbos is the author of backpropagation algorithm. The information must be available to the reader to reproduce the experience, if the reader desire; Information is missing in some references; The proposed method is exhaustively used on the literature; Improve the conclusions (it is very resumed/ there is few points approached on this paper). The ANN used in this paper is considered on the specialized literature, with restricted efficiency Convergence problems, high processing time for the training phase etc.); Thus, I suggest to the Authors to highlight (objectively) the innovation and advantages of this proposal in relation to literature.

Reference

[A]    Werbos, P.J. “Beyond regression: New tools for prediction and analysis in the behavioral sciences, Ph.D. Thesis, Harvard University, 1974.

Author Response

We appreciate your attention to our work and we understand that this process seeks to improve its quality. Follow our comments on the questions proposed:

RC: Reviewer Comment
AC: Authors Comment

RC: English Review: “The present work presents” (?) etc.;

AC: We have corrected such passages in the text. Please, if still some expression of difficult understanding let us know.

RC: Abstract: I suggest to not considering the RMSE value. It is an information with no relevance. The convergence/stability during the training must be emphasized;

AC: We rewrote much of the abstract. Please, see if the contributions of the work are clearer and if the results presented in textual form allow to identify the qualities of the work.

RC: Citations are missing in important parts of the text, for example, some (many) equations, “root mean square error, artificial neural network, support vector machine, decision tree, multilayer perceptron ANN, backpropagation algorithm etc.; I suggest to use benchmark publications, g., backpropagation algorithm [A]. P.J. Werbos is the author of backpropagation algorithm. The information must be available to the reader to reproduce the experience, if the reader desire;

AC: We have included references to complement the text. Please, see the highlighted text (lines 33-36). The unreferenced equations are equations developed by the authors.

RC: Information is missing in some references;

AC: Please, we kindly request which information is missing. All references are complete in our .bib file. However, we may have omitted some setting so that this information is visible.

RC: Improve the conclusions (it is very resumed/ there is few points approached on this paper).

AC: We have tried to include and clarify several points in the conclusion section. Please, see the highlighted text (lines 329-357).

RC: The proposed method is exhaustively used on the literature; The ANN used in this paper is considered on the specialized literature, with restricted efficiency Convergence problems, high processing time for the training phase etc.); Thus, I suggest to the Authors to highlight (objectively) the innovation and advantages of this proposal in relation to literature.

AC: We try to emphasize more the contributions of the work in several passages. Please, see the highlighted text (Abstract; lines 37-42; lines 329-339).

Reviewer 4 Report

It is an interesting paper on the detection of typical motor vibrations. The errors are simulated using a computer cooler fan and several magnets. I have the following comments:

The paper should be put into the fault diagnosis context and not data prediction. The authors should see papers like “Fault Diagnosis for Smart Grids in Pragmatic Conditions” so as to understand the related literature and the usual division between model-based and model-free (cognitive) fault diagnosis methods. Right now the description of the related literature is poor.

The detection method should be formalized. A problem formulation section would help (see  “Model-free fault detection and isolation in large-scale cyber physical systems”).

The experimental protocol and especially the test/train data division should be carried out similarly to the related literature.

The same holds for the figures of merit (fp,fn,dd)

There is no comparison with the state of the art so it is not possible to understand the relevance of this work.

There is no motivation provided behind the use of an ANN for this task.

Author Response

We appreciate your attention to our work and we understand that this process seeks to improve its quality. Follow our comments on the questions proposed:

RC: Reviewer Comment
AC: Authors Comment

RC: The paper should be put into the fault diagnosis context and not data prediction. The authors should see papers like “Fault Diagnosis for Smart Grids in Pragmatic Conditions” so as to understand the related literature and the usual division between model-based and model-free (cognitive) fault diagnosis methods. Right now the description of the related literature is poor.

AC: We try to better contextualize the work in terms of asset management and condition-based maintenance. Fault Diagnosis Systems were addressed in this context. Please, see the highlighted text (lines 54-79).

RC: The detection method should be formalized. A problem formulation section would help (see “Model-free fault detection and isolation in large-scale cyber physical systems”).

AC: The purpose of our work was not to detect the failure, but to predict a future failure to prevent the failure from happening. We try to make the work objectives clearer. Please, see the highlighted text (lines 28-42).

RC: The experimental protocol and especially the test/train data division should be carried out similarly to the related literature. The same holds for the figures of merit (fp,fn,dd)

AC: To perform the trained model test we used the $k$ -fold cross-validation technique, calculating the RMSE performance index. This form of validation of neural network training is standardized in the literature. As the result of prediction is not a form of classification, validation must be performed by indices such as the RMSE. Please, see the highlighted text (lines 256-261).

RC: There is no comparison with the state of the art so it is not possible to understand the relevance of this work.

AC: We try to emphasize more the contributions of the work in several passages. Please, see the highlighted text (Abstract; lines 37-42; lines 329-339).

RC: There is no motivation provided behind the use of an ANN for this task.

AC: We include the justification of the technique used in 2 points in the text. Please, see the highlighted text (lines 81-87; lines 227-247).

Round 2

Reviewer 2 Report

It is recommendable one state of the art to understand the relevance of this work, the used of ANN and the comparison with other techniques or cases.

The application on real industrial cases could improve the relevance of the developed models, this is crucial because locational and operational conditions of the asset could modify the designed ANN. Comparison and sensibility analysis in different real cases will be well accepted.

Author Response

We have tried to include all points commented. We appreciate your attention to our work and we understand that this process seeks to improve its quality. Follow our comments on the questions proposed.

RC: Reviewer Comment
AC: Authors Comment

RC: It is recommendable one state of the art to understand the relevance of this work, the used of ANN and the comparison with other techniques or cases. The application on real industrial cases could improve the relevance of the developed models, this is crucial because locational and operational conditions of the asset could modify the designed ANN. Comparison and sensibility analysis in different real cases will be well accepted.

AC: We included a comparison of the model trained with the ANN technique with models trained with other machine learning techniques (Regression Tree, Random Forest and Support Vector Machine). With this inclusion of content the following changes were made in the manuscript:

- Abstract adjusted to mention which comparisons with other machine learning techniques were made.

- References appropriate for the machine learning techniques compared. Please, see the highlighted text (lines 33-35).

- Comparative tests commented in the introduction section. Please, see the highlighted text (lines 47-50).

- Adjusting in the definition of ANN to emphasize generalization capacity. Please, see the highlighted text (lines 83-89).

- New subsection "Performance Index", to clearly define the technique used to measure the performance of the system. Please, see the highlighted text (lines 222-226).

- Update of the text of the ANN parameters description, to show the parameters in the same way described in other machine learning techniques. Please, see the highlighted text (lines 252-260).

- New subsection "Comparing with other Machine Learning Techniques" to present machine learning techniques that have been compared with ANN (Regression Tree, Random Forest and Support Vector Machine). In this subsection, besides a brief description of each technique, accompanied by the appropriate references, the training parameters of the models were presented (New table 1). Please see the highlighted text (lines 262-308).

- With the inclusion of comparisons between machine learning techniques, it was necessary to rewrite the results section. This section now presents comparative graphs and tables, these results are discussed during the section. Please see the highlighted text (lines 310-364).

- The conclusion was adjusted to comment on the comparative tests. Please see the highlighted text (lines 370-383).

The application of the proposed system in a real environment could not be done in this work, but will be treated in future works as described in the conclusion.

Reviewer 3 Report

-

Author Response

-

Reviewer 4 Report

I would like to thank the authors for considering my comments. I think that the article has substantially improved. However, the paper still lacks a comparison with state of the art methods, e.g. HMMs or another method taken from the related literature.

Author Response

We have tried to include all points commented. We appreciate your attention to our work and we understand that this process seeks to improve its quality. Follow our comments on the questions proposed.

RC: Reviewer Comment
AC: Authors Comment

RC: I would like to thank the authors for considering my comments. I think that the article has substantially improved. However, the paper still lacks a comparison with state of the art methods, e.g. HMMs or another method taken from the related literature.

AC: We included a comparison of the model trained with the ANN technique with models trained with other machine learning techniques (Regression Tree, Random Forest and Support Vector Machine). With this inclusion of content the following changes were made in the manuscript:

- Abstract adjusted to mention which comparisons with other machine learning techniques were made.

- References appropriate for the machine learning techniques compared. Please, see the highlighted text (lines 33-35).

- Comparative tests commented in the introduction section. Please, see the highlighted text (lines 47-50).

- Adjusting in the definition of ANN to emphasize generalization capacity. Please, see the highlighted text (lines 83-89).

- New subsection "Performance Index", to clearly define the technique used to measure the performance of the system. Please, see the highlighted text (lines 222-226).

- Update of the text of the ANN parameters description, to show the parameters in the same way described in other machine learning techniques. Please, see the highlighted text (lines 252-260).

- New subsection "Comparing with other Machine Learning Techniques" to present machine learning techniques that have been compared with ANN (Regression Tree, Random Forest and Support Vector Machine). In this subsection, besides a brief description of each technique, accompanied by the appropriate references, the training parameters of the models were presented (New table 1). Please see the highlighted text (lines 262-308).

- With the inclusion of comparisons between machine learning techniques, it was necessary to rewrite the results section. This section now presents comparative graphs and tables, these results are discussed during the section. Please see the highlighted text (lines 310-364).

- The conclusion was adjusted to comment on the comparative tests. Please see the highlighted text (lines 370-383).

The application of the proposed system in a real environment could not be done in this work, but will be treated in future works as described in the conclusion.

Round 3

Reviewer 2 Report

Due to changes that reinforce the paper, I accept in present form.

Reviewer 4 Report

I  think that the comparison with other ML approaches improved the paper substantially. The paper can be accepted after minor language editing.